# Infliximab for Treatment of Immune Adverse Events and Its Impact on Tumor Response

**DOI:** 10.3390/cancers15215181

**Published:** 2023-10-27

**Authors:** Vishnupriyadevi Parvathareddy, Umut Selamet, Aditi A. Sen, Omar Mamlouk, Juhee Song, Valda D. Page, Maen Abdelrahim, Adi Diab, Noha Abdel-Wahab, Ala Abudayyeh

**Affiliations:** 1Department of Nephrology, Baylor College of Medicine, Houston, TX 77030, USA; ammalu19010@gmail.com (V.P.); asen4@mdanderson.org (A.A.S.); 2Department of Medical Oncology, Dana Farber Cancer Institute, Harvard Medical School, Boston, MA 02215, USA; umut_selamet@dfci.harvard.edu; 3Section of Nephrology, Division of Internal Medicine, The University of Texas MD Anderson Cancer Center, Unit 1468, 1400 Pressler Street, Houston, TX 77030, USA; omamlouk@mdanderson.org (O.M.); vpage@mdanderson.org (V.D.P.); 4Department of Biostatistics, The University of Texas MD Anderson Cancer Center, Houston, TX 77030, USA; jsong1@mdanderson.org; 5Institute of Academic Medicine and Weill Cornell Medical College, Houston Methodist Cancer Center, Houston, TX 77479, USA; mabdelrahim@houstonmethodist.org; 6Department of Melanoma Medical Oncology, The University of Texas MD Anderson Cancer Center, Houston, TX 77030, USA; adiab@mdanderson.org (A.D.); nahassan@mdanderson.org (N.A.-W.); 7Section of Rheumatology and Clinical Immunology, Department of General Internal Medicine, The University of Texas MD Anderson Cancer Center, Houston, TX 77030, USA; 8Rheumatology and Rehabilitation Department, Assiut University Hospitals, Faculty of Medicine, Assiut 71515, Egypt

**Keywords:** acute kidney injury, immune checkpoint inhibitor, cancer progression

## Abstract

**Simple Summary:**

The use of biologic agents in the treatment of immune adverse events (irAEs) due to immune checkpoint inhibitors has been an attractive option but there are limited data on their impact on tumor progression. This study is one of the largest retrospective cohorts that evaluated the predictors of response to infliximab for the treatment of irAEs and infliximab’s impact on tumor response. The study helps to support the safe and effective use of infliximab in treatment of irAEs without significant impact on tumor response.

**Abstract:**

**Background:** Immune-related adverse events (irAEs) challenge the use of immune checkpoint inhibitors (ICIs). We performed a retrospective study to evaluate response to infliximab for immune-related adverse event management, and infliximab’s effect on progression-free survival (PFS) and overall survival (OS) with a focus on melanoma and genitourinary cancers. **Methods:** We retrospectively reviewed records of all cancer patients exposed to infliximab after immune checkpoint inhibitor (ICI) treatment from 2004 to 2021 at the MD Anderson Cancer Center. Survival was assessed utilizing the Kaplan–Meier method. Univariate and multivariate logistic regression was utilized to evaluate predictors of infliximab response, OS, and PFS. **Results:** We identified 185 cancer patients (93 melanoma and 37 genitourinary cancers) treated with ICI and who received infliximab to treat irAEs. Within 3 months of treatment initiation, 71% of the patients responded to infliximab, 27% had no response, and 2% had unknown response. Among different irAEs, colitis was associated with increased response to infliximab at 3 months, irrespective of the type of malignancy. We evaluated best tumor response before and after infliximab in the entire cohort and again in the melanoma and genitourinary (GU); the findings were similar in the melanoma cohort and the entire cohort, where best tumor response before and after infliximab was not significantly different. In the melanoma cohort, acute kidney injury (AKI) was associated with increased risk of death, *p* = 0.0109, and having response to infliximab was associated with decreased risk of death, *p* = 0.0383. Interestingly in GU cancer patients, myositis was associated with increased risk of death, *p* = 0.0041, and having a response to infliximab was marginally associated with decreased risk of death, *p* = 0.0992. As regards PFS, in a multivariate Cox regression model, having a history of cardiovascular disease remained significantly associated with shorter PFS in the melanoma cohort. For patients with GU cancers, response to infliximab was associated with longer PFS. **Conclusions:** Our study is among the largest retrospective analyses of infliximab use for irAE management. Patients with colitis were the best responders to infliximab. AKI before initiation of infliximab in the melanoma subcohort and myositis in GU subcohort are associated with higher risk of death. Our results indicate no association between infliximab and cancer progression with the exception of genitourinary cancers.

## 1. Introduction

Immune check point inhibitors (ICIs) have greatly changed the paradigm of cancer treatment in the past decade. These agents block proteins and ligands, such as cytotoxic T-lymphocyte-associated protein-4 (CTLA-4) and programmed cell death protein 1/programmed cell death ligand 1 (PD-1/PD-L1), that ordinarily inactivate cytotoxic T cells, which are important players of the immune system against tumor cells. Activation of T cells by ICIs can lead to immune-mediated reactions that are similar to autoimmune diseases in almost all organ systems. The immune reactions due to ICIs are referred as immune-related adverse events (irAEs). 

The most common irAEs are observed in the skin, endocrine system, gastrointestinal tract, liver, and lungs, whereas the kidneys and heart are affected less frequently. Since irAEs are the result of an overactive immune system, immunosuppression with corticosteroids is the mainstay of treatment for irAEs. Some studies suggest that irAEs are actually indicators of good tumor response; therefore, corticosteroids may potentially hinder the antitumor effect of ICIs [1]. However, the literature regarding the impact of corticosteroids on tumor response during ICI treatments is conflicting [2]. In addition, the long duration of corticosteroid use leads to steroid-induced adverse events. Therefore, novel immunosuppressive treatments are needed for reversal of irAEs both for steroid-refractory cases and for steroid-sparing purposes. As a result of the challenges associated with irAEs and the interruption of treatment, oncologists and other specialists have started utilizing biological and nonbiological immunosuppressive drugs to treat severe and steroid refractory immune-related adverse events. Extrapolating data from the use of steroid-sparing agents in the setting of inflammatory diseases has led to the increasing use of such agents, especially tumor necrosis factor alpha (TNF-α) blockers and IL-6, in the management of irAEs [3]. The urgency in investigating steroid sparing-agents is further highlighted in dual ICI treatment. For example, in the CheckMate 067 study, melanoma patients receiving combined anti–PD-1 and anti–CTLA-4 therapy led to 52% 5-year overall survival (OS) in the entire cohort, 44% in the nivolumab group, and 26% in the ipilimumab group, but 96% of patients had an associated irAE, including 59% with grade 3–4 irAEs [4]. 

TNF-α is a dual acting cytokine with a proinflammatory effect that plays a crucial role in the development of irAEs and with tumor surveillance properties through its induction of apoptotic cell death in a wide variety of tumor cells [5]. Infliximab, a TNF-α inhibitor, has been extensively utilized to treat steroid-refractory ICI-induced enterocolitis and is being explored for other irAEs [6,7,8,9,10,11,12,13,14,15,16,17,18,19,20,21,22,23,24,25,26]. The overall results of such studies have been promising from the standpoint of irAE resolution, but infliximab’s enhancement of tumor progression was questioned. Preclinical data have demonstrated that TNF-α blockade promotes the infiltration of activated T cells induced by ICI treatment and that combination therapy with anti–CTLA-4 and anti–PD-1 with TNF-α blockade improves tumor responses and decreases irAEs in mouse models of melanoma and colon cancer [24,27,28,29]. Here, we conducted a retrospective study to evaluate the use of infliximab in all irAEs reported at the MD Anderson Cancer Center and its role on irAE response, progression-free survival (PFS), and OS with a focus on melanoma and GU cancer patients.

## 2. Materials and Methods

### 2.1. Study Design and Patient Population

This retrospective study was approved by the institutional review board at The University of Texas MD Anderson Cancer Center, and the procedures followed were in accordance with the principles of the Declaration of Helsinki. Patients were identified by querying MD Anderson pharmacy data from 2004 to 2021, to identify all patients treated with ipilimumab, nivolumab, pembrolizumab, atezolizumab, durvalumab, or avelumab for metastatic cancer or as adjuvant treatment for primary disease who received infliximab for treatment of an irAE. We identified a total of 185 patients with imaging performed before and after infliximab initiation to be included in the analysis.

### 2.2. Data Collection and Definitions

Detailed demographic information for each patient, including age at the initiation of ICI treatment, sex, and race, were attained from MD Anderson’s Epic medical record system. In addition, data on the underlying cancer, types of irAE, steroid exposure prior to infliximab, and indication for infliximab use were collected. Comorbidities identified were cardiovascular disease (CVD), hypertension, diabetes, hyperlipidemia, autoimmune disease, and hypothyroidism. All available serum creatinine values and survival data were collected. Creatinine values were recalculated for the estimated glomerular filtration rate (eGFR) utilizing the Chronic Kidney Disease Epidemiology Collaboration (CKD-EPI) creatinine equation. Baseline eGFR was the first available eGFR prior to ICI initiation and was categorized as stage 1–2 chronic kidney disease (eGFR ≥ 60 cc/min/1.73 m^2^) vs. stage 3–4 chronic kidney disease (eGFR ≤ 60 cc/min/1.73 m^2^) based on the Kidney Disease Improving Global Outcomes (KDIGO) guidelines. 

Acute kidney injury (AKI) was identified by the KDIGO guidelines as an absolute increase in serum creatinine ≥0.3 mg/dL within 48 h. We collected baseline serum creatinine, peak creatinine, and creatinine at 3 months after infliximab exposure.

Tumor response was assessed with best response prior to infliximab and compared to best response after infliximab exposure. This was assessed based on images available before and after infliximab. Overall response was defined as having stable disease, remission, or any improved tumor response after infliximab, and “no response” was defined as any progression of disease. Patients who did not have imaging to assess response before and after infliximab and those who underwent new cancer treatment during the assessment of tumor response were excluded from the analysis. 

Response to infliximab was evaluated from clinical notes based on the irAE indication. IrAE definition and grade were defined based by the common document Terminology Criteria for Adverse Events (CTCAGE) v.5.0. If there was improvement in the irAE, the patient was categorized as a responder, and if the irAE remained the same or worsened, the patient was defined as a non-responder to infliximab.

OS and PFS were evaluated separately in the subgroups of patients with melanoma and GU cancers which were the two major cancers in our cohort.

### 2.3. Statistical Analysis

Patient baseline and demographic characteristics were summarized by descriptive statistics, mean (SD), or median (IQR) for continuous variables and frequency (%) for categorical variables. OS from ICI initiation and OS from infliximab initiation to death were calculated. Patients who were alive at last follow-up were censored at the last follow-up date. PFS was defined as time from infliximab initiation until disease progression or death, whichever occurred first. Those who were alive without progression were censored at the time of response evaluation. Tumor response outcomes (response vs. no response) before and after infliximab were compared utilizing the McNemar test. Univariate logistic regression models were fitted to evaluate the predictors of response to infliximab (response vs. no response within 3 months of infliximab initiation; no response includes patients who had response after 3 months, who had no response with follow-up >3 months, or who died within 3 months). After covariates with a *p*-value less than 0.1 were identified, based on univariate logistic regression models, multivariate logistic regression models were fitted (utilizing the backward elimination method). Univariate Cox regression models were fitted to evaluate the predictors of OS and PFS from infliximab initiation; melanoma patients and GU cancer patients were evaluated separately. Multivariate Cox regression models were explored including ICI type and other covariates presenting a significant or a marginally significant effect (*p*-value < 0.1) based on univariate Cox regression models. Cox regression models with time-varying covariates (AKI and response) were also fitted. A *p*-value < 0.05 indicated statistical significance. SAS 9.4 (SAS Institute, Cary, NC, USA) was utilized for data analysis. 

## 3. Results

We identified and analyzed 185 patients who were diagnosed with cancer and treated with ICIs and, subsequently, developed an irAE that was treated with infliximab over the period studied. Patient characteristics are summarized in Table 1. The median follow-up time was 36.2 months (95% CI by reverse Kaplan–Meier, 30.6–42.8). The median time from ICI initiation to irAE was 2.4 months (IQR, 1.4–6.0) and the median time from irAE to infliximab initiation was 8 days (IQR, 4–23). Of the entire cohort, 73% were male and 27% were female, with a median age of 64 years. More than half (58.9%) of the patients received concurrent treatment with two ICIs while 41.1% received single-agent ICIs; 30.8% had been exposed to steroids prior to infliximab. Hypertension was the highest reported comorbidity at 63.2%. A total of 7% of the patients had autoimmune disease at baseline and 38.4% had AKI during the study period with 23.8% with AKI prior to infliximab and 30.8% with AKI within 1 month of infliximab treatment. The most common underlying cancer was melanoma (50% of the study cohort), followed by GU cancers (20%). 

The most common irAE reported and treated with infliximab was colitis, followed by pneumonitis, acute interstitial nephritis, myocarditis, and myositis (Figure 1). A higher rate of colitis was noted in patients treated with combination ICIs (72.5%) compared to those treated with monotherapy, and this finding was consistent in the melanoma and GU subcohorts (Appendix A). For all irAEs, steroids were less likely to be utilized prior to infliximab (30.8%) and were more commonly given before infliximab in patients treated with ICI monotherapy (*p* < 0.0001). Among patients treated with ICI monotherapy, all six patients with myocarditis received steroids prior to infliximab, whereas 32 of the 70 patients without myocarditis received steroids prior to infliximab (*p* = 0.0253). There was no significant association between irAE and steroid use prior to infliximab in patients receiving combination ICI treatment.

### 3.1. Predictors of Response to Infliximab

Predictors of response to infliximab within 3 months of its initiation for the entire cohort was evaluated, and 132 patients had a response within 3 months of treatment compared to 50 with no response (3 patients had unknown response). ICI type showed a marginal significance (0 = 0.0569) based on univariate model (Appendix A). In a multivariate model, patients with colitis were correlated with increased odds of having response (OR 4.59, 95% CI 2.275–9.262, *p* < 0.0001: Figure 2), while ICI type was not significantly associated. Development of AKI before and within 1 month after infliximab initiation were not associated with response to infliximab (Appendix A). In the melanoma and GU cancer patient subgroups, multivariate analysis again showed that development of colitis as an irAE was significantly associated with response to infliximab at 3 months (melanoma, OR 3.242, 95% CI 1.072–9.806, *p* = 0.0373; GU cancer, OR 8.736, 95% CI 1.482–51.475; *p* = 0.0166), while ICI type was not significantly associated.

### 3.2. Best Tumor Response Assessment 

Best tumor response before and after infliximab initiation were compared for all patients and for the subgroups of melanoma patients and GU cancer patients (Table 2, Figure 3a–c). Of the total of 185 patients, 141 had images before and after infliximab and had not started new cancer treatment during the assessment of cancer response. The marginal distribution of best tumor response prior to infliximab initiation and the marginal distribution of best tumor response after infliximab initiation were not different for all patients (*p* = 0.0961, Figure 3a) or for melanoma patients (*p* = 0.5775, Figure 3b) but were marginally significant for GU cancer patients (*p* = 0.0339, Figure 3c). This indicated that patients treated for an irAE with infliximab were not more likely to develop cancer progression after infliximab exposure, with the exception of patients with GU cancers. 

### 3.3. Progression-Free Survival

PFS was assessed in patients with melanoma and GU cancers separately. PFS was defined as time from infliximab initiation until disease progression or death, whichever occurred first. Those who were alive without progression (based on response evaluation after infliximab initiation) or death were censored at the time of response evaluation. Median PFS was 9.7 months (95% CI 6.7–17.6) for melanoma patients and 4.5 months (95% CI 3.2–11.7) for GU cancer patients (Figure 4a–c).

In the melanoma cohort, CVD (*p* < 0.0001) was significantly associated with shorter PFS based on univariate model (Appendix A). In a multivariate Cox regression model, CVD remained significantly associated with shorter PFS, (HR 3.776, 95% CI 1.907–7.475, *p* = 0.0001; Figure 5). We also fitted a Cox regression model with time-varying covariates (AKI status and response to infliximab) and, again, CVD remained significant. For patients with GU cancers, no variables were significantly associated with PFS based on univariate Cox models (Appendix A). We also fitted a Cox regression model with time-varying covariates (AKI status and response to infliximab). Only response to infliximab was associated with longer PFS (Table 3).

### 3.4. Overall Survival 

Median OS was 29.4 months (95% CI 19.3–39.7) for the entire cohort. For melanoma patients, median OS was 42.3 months, and for GU cancer patients, median OS was 33.1 months (Figure 6a–c). Among melanoma patients, factors associated with OS from infliximab initiation are presented in Appendix A. Univariate Cox models identified steroids before infliximab (*p* = 0.10), CVD (*p* = 0.05), and AKI prior to infliximab (*p* = 0.044) as marginally significant. AKI prior to infliximab remained a significant predictor of poor survival in the melanoma cohort (HR 2.113, 95% CI 1.005–4.441; *p* = 0.0485, Figure 7a). We also fitted a Cox regression model with time-varying covariates (AKI status and response to infliximab). Having AKI (HR 2.269. 95% CI 1.208–4.262, *p* = 0.0109) was associated with increased risk of death and having response to infliximab was associated with decreased risk of death (HR 0.470, 95% CI 0.230–0.960, *p* = 0.0383; Table 4).

Among patients with GU cancers, factors associated with OS from infliximab initiation are presented in Appendix A. Univariate Cox models identified myocarditis (*p* = 0.0677) as marginally significant and myositis (*p* = 0.0005) as significant. In a multivariate model, myositis remained significantly associated with increased risk of death (Figure 7b). 

We also fitted a Cox regression model with time-varying covariates (AKI status and response to infliximab). Having AE myositis was associated with increased risk of death (HR 7.637, 95% CI 1.907–30.590, *p* = 0.0041), and having a response to infliximab was marginally associated with decreased risk of death (HR: 0.334, 95% CI 0.091–1.230, *p* = 0.0992; Table 4).

## 4. Discussion 

In this large, retrospective study, we analyzed records of 185 cancer patients (including 93 melanoma patients) who received ICI for cancer treatment and infliximab for treating various irAEs. We found that patients with ICI-induced colitis were the best responders to infliximab. Infliximab use has been widely recognized for steroid-refractory cases of ICI-induced colitis given its known efficacy against inflammatory bowel disease, which is considered the autoimmune counterpart of ICI-induced colitis [6]. There has also been increasing evidence regarding the use of TNF blockade in other irAEs such as pneumonitis and nephritis [10,30].

Also, in our study, AKI before the initiation of infliximab was associated with worse OS rates in the melanoma subcohort. Similar to our finding of an association of AKI with death, García-Carro et al. showed that, in 759 cancer patients treated with ICI, a single episode of AKI was independently associated with increased mortality [31]. However, another study from our group found no association between AKI and OS in a cohort of 1664 melanoma patients treated with ICIs [32]. Recently, Baker et al. showed that AKI was independently associated with mortality in 2207 patients who received ICIs, but mortality was higher if AKI was unrelated to ICI use and lower if AKI was due to acute interstitial nephritis, which is the main pathology seen with ICI-induced AKI [33]. In our study, AKI events that were observed before infliximab initiation were most likely unrelated to ICI use, as only 5.4% of all reported irAEs were acute interstitial nephritis, whereas 23.8% of all patients had AKI prior to infliximab initiation. 

Our study suggests that infliximab use had no meaningful impact on cancer progression in the melanoma patients. TNF-α plays a pleiotropic role in cancer. On one hand, it promotes inflammation, which is the driving force for irAEs; on the other hand, it induces apoptotic cell death of cancer cells [5]. Given TNF-α’s role in tumor surveillance, it is natural to expect TNF-α blockade to negatively affect cancer progression. Few studies have analyzed the effect of infliximab use on cancer progression. Zou et al. studied efficacy and safety of vedolizumab and infliximab treatment for immune-mediated diarrhea and colitis in 184 cancer patients and showed that all three treatment groups (vedolizumab monotherapy, infliximab monotherapy, and vedolizumab plus infliximab combination therapy) had increased cancer progression, but the infliximab monotherapy group had the highest rate of cancer progression [26]. Both our study and the study by Zou et al. included patients with melanoma and GU cancers and had similar sample sizes. However, our study analyzed heterogenous irAEs, whereas the study by Zou et al. included only immune colitis; therefore, the durations of steroid exposure prior to infliximab were likely different in each study, accounting for additional risk factors for cancer progression. Infliximab use has been effective in resolving ICI-induced colitis without having an impact on tumor responses [6,34,35]. However, a retrospective Dutch cohort of 1250 patients with unresectable locally advanced melanoma treated with ICIs suggested that management of irAEs utilizing a combination of infliximab and corticosteroids was associated with shorter OS compared with corticosteroids alone, although the authors commented that a limitation of their study was that most of the anti-TNF–treated cohort also received high-dose steroids [36]. Interestingly, there has been preclinical evidence that TNF-α blockade can promote tumor regression and enhance ICI-mediated T cell activation against the cancer cells [24,27]. Based on this finding, a recent clinical trial by Montfort et al. (NTC03293784) evaluated 14 patients by studying two separate cohorts utilizing certolizumab and infliximab (Anti-TNF) in addition to combining Nivolumab/ipilimumab. The study concluded that the combination of anti-TNF was safe and did not impair tumor response. In addition, activation and maturation of systemic T cell responses were seen in patients from both cohorts, and indications of reorientation of the systemic immune response toward Th1 responses was believed to be accentuated by the ICI [37]. Based on its promising results, the study is expanding recruitment. Similar to TNF-α blockade, there has been increasing interest in the use of an IL-6 inhibitor (tocilizumab) as another attractive option with proven effectiveness for treatment of irAEs without having an impact on tumor response [3,38,39]. Several clinical trials are underway to investigate the use of IL-6 blockade in the treatment of irAEs (NCT04940299, NCT04375228, NCT03601611, NCT03999749, NCT03821246, NCT04524871, NCT03424005, and NCT03869190). 

Our study has several limitations. This is a retrospective and single-center study, so our results cannot be generalized. We were not able to collect all data retrospectively on cancer stages and different cancer treatments. The heterogeneity of irAEs confounds the analysis of response categorization. Also, our study consisted of both melanoma and GU cancer patients, so different tumor biology confounds the cancer PFS and OS analyses for the entire group, which we tried to overcome with separate analysis of each subgroup. 

## 5. Conclusions

In this study we presented the largest retrospective analysis of infliximab use for irAE management. We found that colitis was the irAE type with the most responses to infliximab. AKI within 1 month of infliximab use hampered irAE response to infliximab. AKI before initiation of infliximab in melanoma patients was associated with worse OS. There were more patients with cancer progression after infliximab initiation compared to before infliximab, but this finding was confounded by the natural time course of cancer. Interestingly, most patients who were in remission before infliximab initiation remained in remission after infliximab, suggesting no meaningful association of infliximab on cancer progression in the melanoma patients. Given the limitations of our study owing to its retrospective and single-center characteristics, large-scale, multicenter, prospective studies are needed for assessing the safety and efficacy of infliximab for treatment of irAEs. 

## Figures and Tables

**Figure 1 cancers-15-05181-f001:**
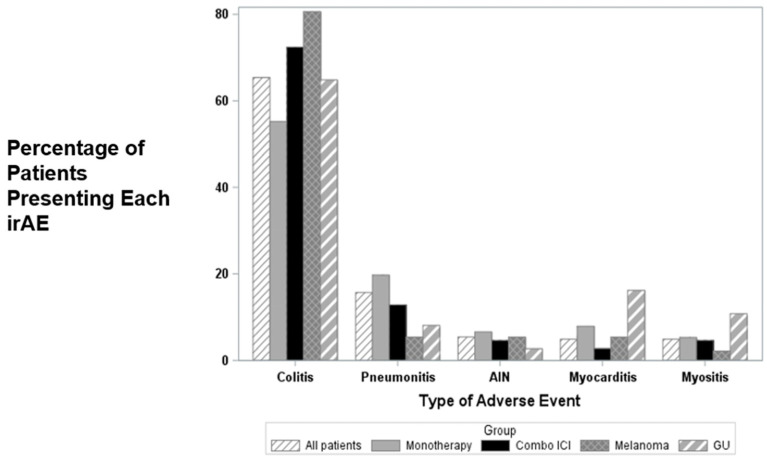
Percentage of patients presenting each irAE in all subgroups of therapy and cancers for 5 most frequent AEs. AIN, acute interstitial nephritis; GU, genitourinary.

**Figure 2 cancers-15-05181-f002:**
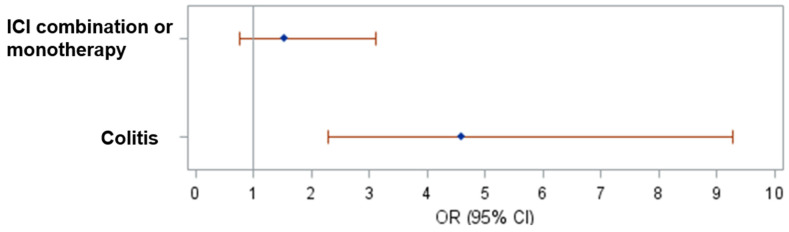
Forest plot of the odds ratios and their 95% CIs based on multivariate logistic regression model.

**Figure 3 cancers-15-05181-f003:**
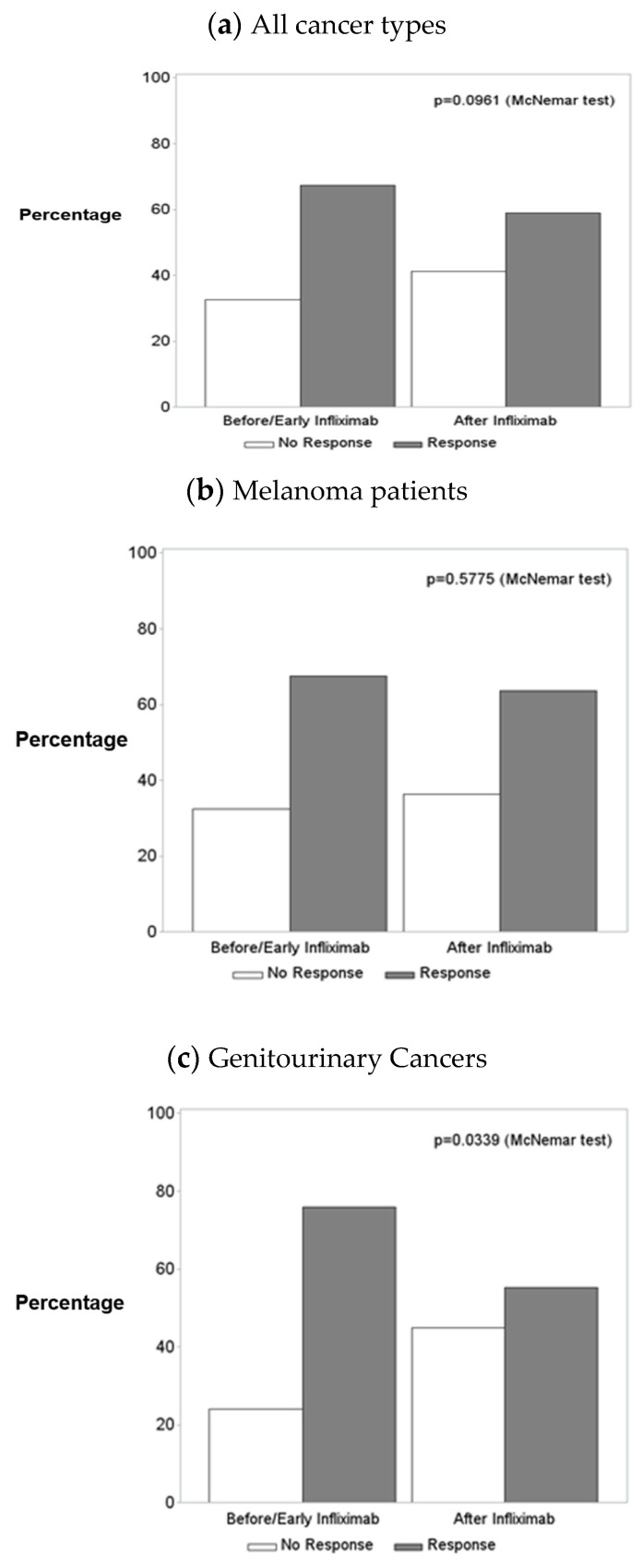
Best tumor response prior to infliximab initiation vs. best tumor response after infliximab initiation for (**a**) all patients, (**b**) melanoma patients, and (**c**) genitourinary cancer patients.

**Figure 4 cancers-15-05181-f004:**
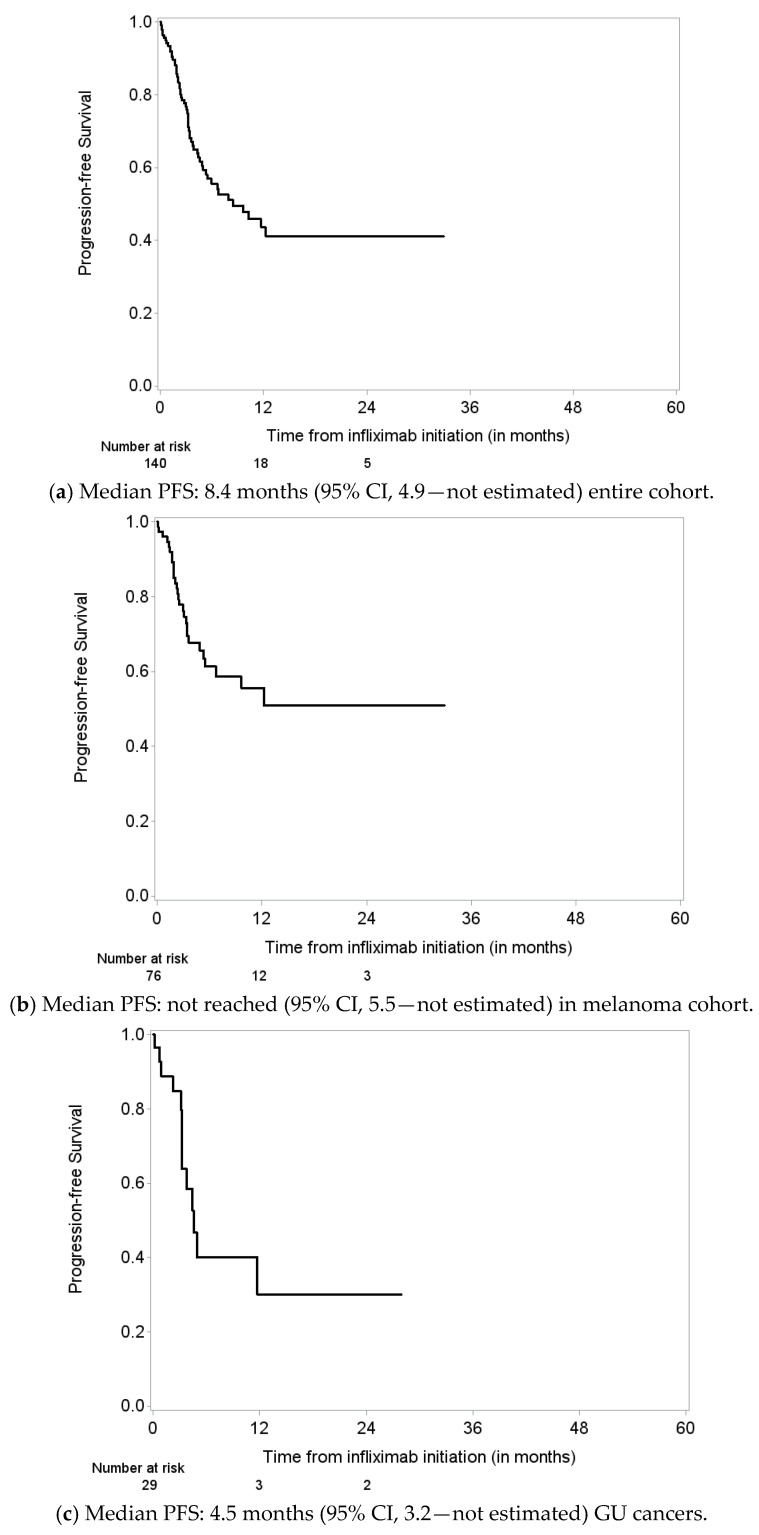
Progression-free survival (PFS) from infliximab initiation until disease progression or death; those alive without progression were censored at the time response evaluation in (**a**) all patients, (**b**) melanoma patients, and (**c**) genitourinary cancer patients.

**Figure 5 cancers-15-05181-f005:**
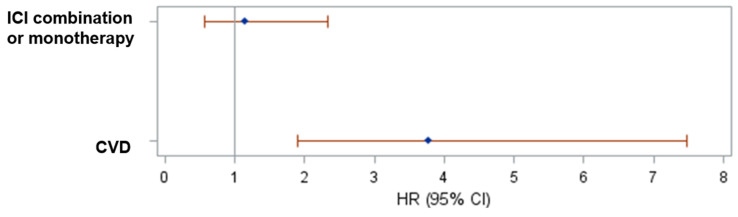
Forest plot of the hazard ratios and their 95% CIs of progression or death based on multivariate Cox regression model including melanoma patients.

**Figure 6 cancers-15-05181-f006:**
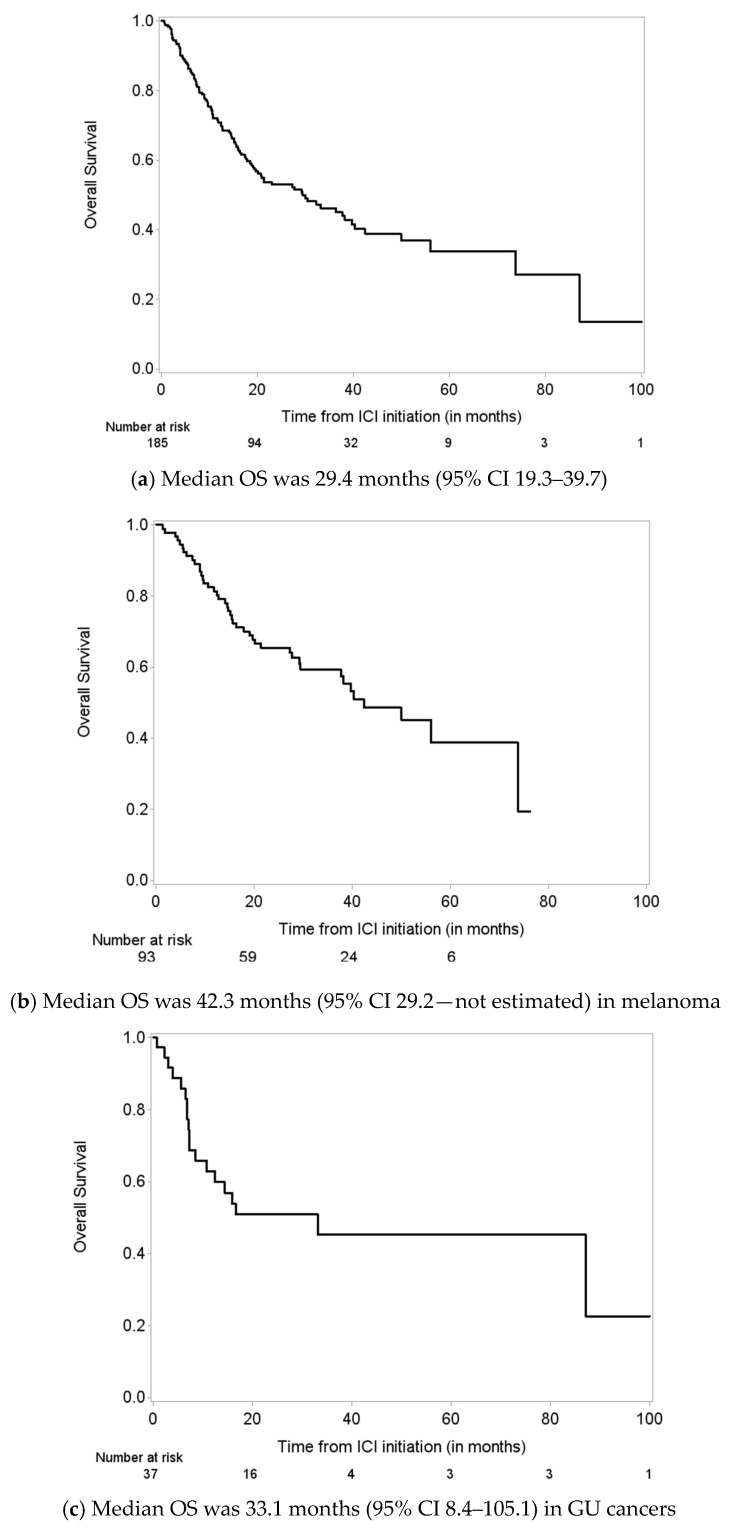
Overall survival (OS) from immune checkpoint inhibitor (ICI) initiation for (**a**) all patients, (**b**) melanoma patients, and (**c**) genitourinary cancer patients.

**Figure 7 cancers-15-05181-f007:**
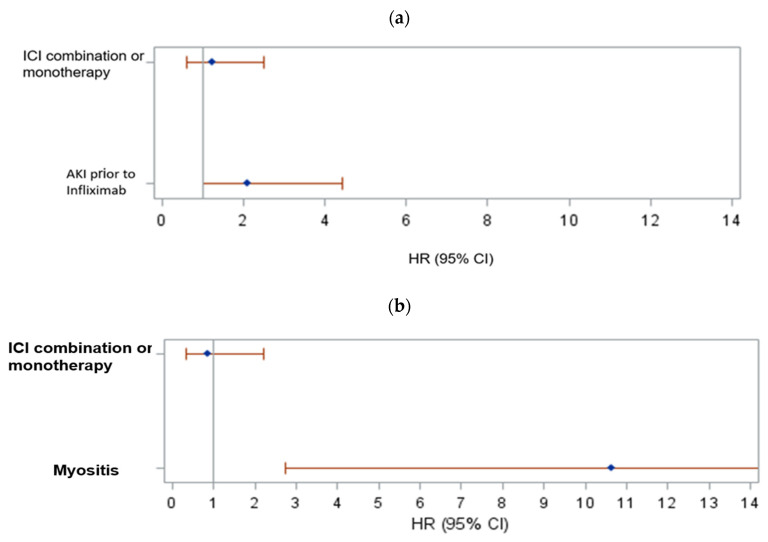
(**a**) Forest plot of the hazard ratios and their 95% CIs of death based on multivariate Cox regression model including melanoma patients; (**b**) forest plot of the hazard ratios and their 95% CIs of death based on multivariate Cox regression model including GU patients.

**Table 1 cancers-15-05181-t001:** Patient demographic and baseline characteristics for melanoma and genitourinary cancer patients.

Cancer Type		Melanoma (n = 93)	Genitourinary (n = 37)	All (n = 185)
Covariate	Level	Descriptive statistics	Descriptive statistics	Descriptive statistics
Age at ICI initiation (years)	Median (IQR)	62.99 (49.81–71.12)	69.94 (58.33–75.61)	63.93 (55.62–73.08)
Duration of ICI (months)	Median (IQR)	4.3 (1.64–12.58)	1.38 (0.95–2.79)	2.23 (1.02–8.25)
Duration of infliximab (days)	Median (IQR)	1 (1–35)	20 (12–45)	1 (1–34)
Number of infliximab doses	Median (IQR)	1 (1–2)	2 (2–3)	1 (1–2)
Baseline creatinine	Median (IQR)	0.88 (0.77–1.01)	1 (0.8–1.3)	0.9 (0.77–1.02)
Time from ICI to irAE (months)	Median (IQR)	2.76 (1.54–7.69)	2.23 (1.15–3.94)	2.43 (1.38–6.05)
Time from irAE to infliximab (days)	Median (IQR)	7 (3–27)	7 (4–19)	8 (4–23)
Sex	Female	32 (34.4%)	8 (21.6%)	50 (27%)
	Male	61 (65.6%)	29 (78.4%)	135 (73%)
ICI type	Monotherapy	24 (25.8%)	19 (51.4%)	76 (41.1%)
	Combination	69 (74.2%)	18 (48.6%)	109 (58.9%)
	Unknown	1 (1.1%)	0 (0%)	1 (0.54%)
Dialysis	No	89 (95.7%)	36 (97.3%)	177 (96.2%)
	Yes	3 (3.2%)	1 (2.7%)	7 (3.8%)
Steroids before infliximab	No	67 (72%)	26 (70.3%)	128 (69.2%)
	Yes	26 (28%)	11 (29.7%)	57 (30.8%)
HTN	No	39 (41.9%)	8 (21.6%)	68 (36.8%)
	Yes	54 (58.1%)	29 (78.4%)	117 (63.2%)
DM	No	69 (74.2%)	24 (64.9%)	136 (73.5%)
	Yes	24 (25.8%)	13 (35.1%)	49(26.5%)
HLD	No	67 (72%)	23 (62.2%)	123 (66.5%)
	Yes	26 (28%)	14 (37.8%)	62 (33.5%)
CVD	No	77 (82.8%)	31 (83.8%)	149 (80.5%)
	Yes	16 (17.2%)	6 (16.2%)	36 (19.5%)
Hypothyroid	No	59 (63.4%)	29 (78.4%)	130 (70.3%)
	Yes	34 (36.6%)	8 (21.6%)	55 (29.7%)
Autoimmune disease	No	86 (92.5%)	37 (100%)	172 (93%)
	Yes	7 (7.5%)	0 (0%)	13 (7%)
AKI any time	No	67 (72%)	19 (51.4%)	114 (61.6%)
	Yes	26 (28%)	18 (48.6%)	71 (38.4%)
AKI prior to infliximab	No	80 (86%)	23 (62.2%)	141 (76.2%)
	Yes	13 (14%)	14 (37.8%)	44 (23.8%)
AKI within 1 month of infliximab	No	75 (80.6%)	22 (59.5%)	128 (69.2%)
	Yes	18 (19.4%)	15 (40.5%)	57 (30.8%)

AKI, acute kidney injury; CVD, cardiovascular disease; DM, diabetes mellitus; HLD, hyperlipidemia; HTN, hypertension; GU, genitourinary; ICI, immune checkpoint inhibitor; irAE, immune-related adverse even.

**Table 2 cancers-15-05181-t002:** McNemar test comparing the marginal distribution of best tumor response prior to infliximab initiation and the marginal distribution of best tumor response after infliximab initiation, including those with both pre- and post-infliximab tumor response, excluding those receiving other cancer treatments after stopping infliximab.

Best Tumor Response before Infliximab	Response after Infliximab, n (%)		Total, n (%)
	No Response	Response	
No response	26	20	46 (32.62%)
Response	32	63	95 (67.38%)
Total	58 (41.13%)	83 (58.87%)	141

**Table 3 cancers-15-05181-t003:** Cox regression model with time-varying covariates for AKI and response to infliximab for PFS.

Covariate	Level	Melanoma				Genitourinary			
		Univariate Cox Model		Multivariate Cox Model		Univariate Cox Model		Multivariate Cox Model	
		HR (95% CI)	*p*-Value	HR (95% CI)	*p*-Value	HR (95% CI)	*p*-Value	HR (95% CI)	*p*-Value
ICI type	Monotherapy	1.000				1.000		1.000	
	Combination	1.404 (0.711–2.773)	0.3289	1.182 (0.584–2.391)	0.6427	1.250 (0.533–2.934)	0.6077	1.921 (0.759–4.865)	0.1683
CVD	No	1.000				1.000			
	Yes	3.897 (1.998–7.601)	<0.0001	3.475 (1.735–6.956)	0.0004	0.539 (0.235–1.239)	0.1458		
AKI	No	1.000				1.000			
	Yes	1.755 (0.956–3.221)	0.0696	1.503 (0.812–2.782)	0.1949	1.110 (0.473–2.608)	0.8104		
Response	No	1.000				1.000		1.000	
	Yes	0.662 (0.349–1.255)	0.2061			0.468 (0.190–1.155)	0.0996	0.246 (0.096–0.627)	0.0033

AKI, acute kidney injury; CVD, cardiovascular disease; ICI, immune checkpoint inhibitor.

**Table 4 cancers-15-05181-t004:** Cox regression model with time-varying covariates for AKI and response to infliximab for OS.

Covariate	Level	Melanoma				Genitourinary			
		Univariate Cox Model		Multivariate Cox Model		Univariate Cox Model		Multivariate Cox Model	
		HR (95% CI)	*p*-Value	HR (95% CI)	*p*-Value	HR (95% CI)	*p*-Value	HR (95% CI)	*p*-Value
ICI type	Monotherapy	1.000		1.000		1.000		1.000	
	Combination	1.266 (0.622–2.579)	0.5158	1.413 (0.679–2.940)	0.3549	0.741 (0.292–1.880)	0.5282	1.320 (0.422–4.133)	0.6331
CVD	No	1.000							
	Yes	1.989 (0.997–3.969)	0.0511						
Myositis	No					1.000		1.000	
	Yes					10.978 (2.871–41.978)	0.0005	7.637 (1.907–30.590)	0.0041
AKI	No	1.000		1.000		1.000			
	Yes	2.630 (1.428–4.842)	0.0019	2.269 (1.208–4.262)	0.0109	0.788 (0.303–2.046)	0.6241		
Response	No	1.000		1.000		1.000		1.000	
	Yes	0.414 (0.209–0.818)	0.0112	0.470 (0.230–0.960)	0.0383	0.283 (0.103–0.777)	0.0144	0.334 (0.091–1.230)	0.0992

AKI, acute kidney injury; CVD, cardiovascular disease; ICI, immune checkpoint inhibitor.

## Data Availability

All data generated or analyzed during this study are included in this published article.

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
