# Peer review of "Infliximab for Treatment of Immune Adverse Events and Its Impact on Tumor Response"

_cancers, 2023, doi:10.3390/cancers15215181_

Round 1

Reviewer 1 Report

Comments and Suggestions for Authors

dear authours 

the paper is the largest experience on infliximab use for irAEs

I appreciate your effort

I would integrate more information about the cancer prognosis stage and line of treatment if possible.

Author Response

We appreciate the reviewer's kind comments and interest in our work. Since this is a large retrospective study, it is difficult to collect all the data on stages of cancers changes in cancer treatments manually for these patients. We agree this would strengthen the study and a prospective one is being planned currently to collect all relevant data you have requested.

Reviewer 2 Report

Comments and Suggestions for Authors

Abstract/Background:

  1. Define AKI before using it as an abbreviation in line 38. 
  2. The conclusion that the results indicate no association between infliximab and cancer progression may be misleading. The authors may want to include the caveat "with the exception of genitourinary cancers" as we see in Fig 1 that after infliximab there are more individuals with "no response".

Introduction:

  1. Line 89: reference can be done as 6-26, instead of 6-25,26

Materials and Methods:

  1. Line 113: Sentence that starts with "In addition, data the on underlying" is grammatically incorrect. "In addition, data on the underlying."
  2. Line 155: Why did the authors include covariates as those with a P-value cutoff of 0.1? Can the authors give some example covariates here to help clarify?

Results:

  1. For Table 1b, the authors should consider converting the information into a stacked bar graph faceted by patient/treatment type (GU, melanoma, all, mono, combo). Although the information is useful, it is not easy to see the distribution of the data by listing many values in the tables. This makes it difficult for reader interpretability. The original table can be supplementary.
  2. Table 2 requires more explanation. For example, how is ICI combo tested in the univariate logistic model vs. the multivariate logistic model? When testing combo ICI, what are your predictors in the univariate logistic model vs. your multivariate logistic model? The reader will benefit from some further description and interpretation.
  3. Please move the Tables as supp tables as they hinder the readability. To retain the information provided in these tables, please plot the HR in the Tables 2, 4, 5 as a forest plot. This will considerably help the reader to absorb the information. These tables are difficult to read with too many rows and numbers.
  4. Figure 1: Why are the x-axis labels before and after "TNF-alpha". I believe this is referring to before and after infliximab treatment, as the description suggests. If so, the label should either say "TNF-alpha inhibitor" or infliximab.
  5. Figure 2: Please clarify if the x-axis of these plots are showing time after infliximab initiation, as stated in line 243, or time after ICI initiation, as stated in the Figure 2 description. 
  6. Line 283: Should say Fig 3b not Fig 2b.
  7. Figure 3a: Please define/describe CPI. Is this supposed to say ICI?

Discussion and Conclusion:

  1. Line 344: The authors state that based on their findings, infliximab had no meaningful impact on cancer progression. However in the main text, we see that we see in Fig 1c that the percent of individuals with no tumor response increases after infliximab initiation. This should be discussed and addressed in the discussion section.
  2. Line 392: "confounded" misspelled 
Comments on the Quality of English Language

The quality of English is good with some minor revisions required. See "Comments and Suggestions for Authors
".

Author Response

We appreciate the reviewer's kind comments and  interest in our work. Please see the attachment.

Reviewer 3 Report

Comments and Suggestions for Authors

Thank you for conducting this valuable and necessary retrospective review. The study design is well described. In the survival/PFS analysis, I missed differentiation between metastatic and adjuvantly treated patients. In the response analysis, patients who had started new treatment during the analysis period, were not included. Is information available on why they started new treatment? In how many cases was this due to clinically progressive disease? 

In Table 1.a, I think an error was made in 'Time from ICI to irAE': it says it is expressed in days, but I believe it is expressed in months. Given the median time from ICI to irAE being 2.76 months, is it usual in your center to have performed response analysis prior to this time? 

I was intrigued by your conclusion that because patients who were in remission prior to infliximab, usually remained in remission afterwards, you suggest no meaningful association of infliximab on cancer progression. Most concern about infliximab is in the early treatment period, where responses still need to happen, and in Table 3 I read that only 32% of the patients who did not respond prior to infliximab treatment, responded afterwards. I believe including a parallel control group with irAEs who were NOT treated with infliximab, would provide more information on the antitumoral safety.

Comments on the Quality of English Language

The level of English is high, but some sentences were not clearly constructed. The message from line 183-184 was not clear to me. Line 244 "Those who experienced disease progression or died were censored the earlier of progression or death" also did not make sense. 

Author Response

We appreciate the reviewer's kind comments and interest in our work. Please see the attachment for our replies to the comments,
